# Vitamin K: Double Bonds beyond Coagulation Insights into Differences between Vitamin K1 and K2 in Health and Disease

**DOI:** 10.3390/ijms20040896

**Published:** 2019-02-19

**Authors:** Maurice Halder, Ploingarm Petsophonsakul, Asim Cengiz Akbulut, Angelina Pavlic, Frode Bohan, Eric Anderson, Katarzyna Maresz, Rafael Kramann, Leon Schurgers

**Affiliations:** 1Division of Nephrology, RWTH Aachen University, 52074 Aachen, Germany; mhalder@ukaachen.de (M.H.); rkramann@ukaachen.de (R.K.); 2Department of Biochemistry, Cardiovascular Research Institute Maastricht, 6200MD Maastricht, The Netherlands; p.petsophonsakul@maastrichtuniversity.nl (P.P.); a.akbulut@maastrichtuniversity.nl (A.C.A.); a.pavlic@maastrichtuniversity.nl (A.P.); 3NattoPharma ASA, 0283 Oslo, Norway; frode@nattopharma.com (F.B.); eric.anderson@nattopharma.com (E.A.); 4International Science & Health Foundation, 30-134 Krakow, Poland; katarzyna.maresz@nutricon.eu

**Keywords:** vitamin K1, vitamin K2, vitamin K dependent proteins, vascular calcification

## Abstract

Vitamin K is an essential bioactive compound required for optimal body function. Vitamin K can be present in various isoforms, distinguishable by two main structures, namely, phylloquinone (K1) and menaquinones (K2). The difference in structure between K1 and K2 is seen in different absorption rates, tissue distribution, and bioavailability. Although differing in structure, both act as cofactor for the enzyme gamma-glutamylcarboxylase, encompassing both hepatic and extrahepatic activity. Only carboxylated proteins are active and promote a health profile like hemostasis. Furthermore, vitamin K2 in the form of MK-7 has been shown to be a bioactive compound in regulating osteoporosis, atherosclerosis, cancer and inflammatory diseases without risk of negative side effects or overdosing. This review is the first to highlight differences between isoforms vitamin K1 and K2 by means of source, function, and extrahepatic activity.

## 1. Introduction

Vitamin K was first identified in 1936 to be a key factor in blood clotting. When chickens were fed a low-fat diet, they exhibited significantly lower coagulation capacity, resulting in severe bleeding [1]. The lipid fraction of diet was analyzed, and a novel antihemorrhagic factor was discovered. This lipid soluble factor was given the first letter in the alphabet available which coincided with the first letter of the German word “Koagulation” and deemed to be only essential for its anti-hemorrhagic trait [1]. Since then, non-coagulant functions have been discovered and have attracted research interest in several fields around the world. The vitamin K family is comprised of multiple similarly structured fat-soluble molecules containing a 2-methyl-1,4-naphthoquinone ring structure called menadione. Menadione (K3) is of synthetic origin, though due to reported adverse effects of hemolysis and liver toxicity will not encompass the scope of this review [2,3,4]. Naturally, vitamin K occurs as two vitamers: vitamin K1 (also known as phylloquinone) and vitamin K2 (designated also as menaquinones (MKs)). Phylloquinone contains a phytyl side chain which comprises 4 prenyl units [5]. Menaquinones contain an unsaturated aliphatic side chain with a variable number of prenyl units. The number of prenyl units indicates the respective type of menaquinone. Vitamin K2 can be divided into subtypes, namely, short-chain (i.e., menaquinone-4; MK-4) and long-chain (i.e., MK-7, MK-8, and MK-9). For K2, no official reference daily intake (RDI) exists at present. Nevertheless, the effects of K2 on improving health in cardiovascular disease (CVD), bone metabolism, chronic kidney disease and particular cancers has been subject to research in the last decades. The aim of this review is to signify and elaborate on the differences between K1 and K2, by source, and function with an emphasis on non-coagulant mechanisms of vitamin K.

## 2. Dietary Vitamin K

Vitamin K1 is the predominant form of vitamin K present in the diet [6,7]. K1 is predominantly found in green vegetables and plant chlorophylls, whereas K2 menaquinones are synthesized by bacteria [8] and are primarily found in food where bacteria are part of the production process [5,9]. Major sources of K1 include spinach, cabbage, and kale, and absorption of dietary K1 is increased in presence of butter or oils. Beyond leafy greens, K1 can also be found in fruits like avocado, kiwi and grapes [10,11]. The main known sources of K2 are fermented food, meat, and dairy produce [12] (Figure 1). Fermentation of soy beans with *Bacillus natto* produces Natto, a Japanese dish that contains the highest content of K2, in particular MK-7 (321 ng/g of K1, and 10,985 ng/g of K2) [13]. Dairy products are the second richest source of K2 in the diet. Hard cheeses are considered to have the highest amount of menaquinones [14]. Other notable sources of K2 are chicken meat, egg yolks, sauerkraut, beef and salmon [12] (Figure 1).

### 2.1. Vitamin K in Vegetables

One of the best representatives in this group, containing both forms of vitamin K, is sauerkraut (22.4 μg per 100 g of K1, and 5.5 μg per 100 g of K2) [14]. Leafy green vegetables show the highest amount of vitamin K1. Vitamin K1 was present in collards (706 μg per 100 g), in turnip (568 μg per 100 g), spinach (96.7 μg per 100 g), kale (75.3 μg per 100 g), broccoli (146.7 μg per 100 g), soybeans roasted (57.3 μg per 100 g), and carrot juice (25.5 μg per 100 g) [7,15,16].

### 2.2. Vitamin K in Fruits and Nuts

It has been shown by a US-led investigation that fruits and nuts do not generally contain K1, with the exception of kiwifruit (33.9–50.3 μg per 100 g), avocado (15.7–27.0 μg per 100 g), blueberries (14.7–27.2 μg per 100 g), blackberries (14.7–25.1 ug per 100 g), grapes red and green (13.8–18.1 μg per 100 g), dried figs (11.4–20.0 μg per 100 g) and dried prunes (51.1–68.1 μg per 100 g). K1 was present in several nuts during this study; pine nuts (33.4–73.7 μg per 100 g), cashews (19.4–64.3 μg per 100 g), and pistachios (10.1–15.1 μg per 100 g) [10]. Other fruit and nuts reported in the study contain vitamin K1 in insignificant traces. Further to this, vitamin K from fruit and nuts in the diet does not interfere with anticoagulation therapy in patients on warfarin [10].

### 2.3. Vitamin K in Cheese

Vitamin K content in cheese varies depending on a range of factors in production, such as time of ripening and regional differences. These dictate not only the type of cheese but fat and nutrient content. Typically, Dutch hard cheeses contain more K2 compared to softer Mediterranean cheeses. This is most likely influenced by duration of the fermentation process and the nature of bacterial strains used [14]. Although none of these cheeses can be considered an individual source of vitamin K2, consumption can contribute to total vitamin K levels [17]. Vitamin K1 and K2 were assessed in European cheeses and highest amount of K1 was found in Roquefort (6.56 μg per 100 g), Pecorino (5.56 μg per 100 g), Brie (4.92 μg per 100 g), Boursin (4.55 μg per 100 g), Norvegia (4.37 μg per 100 g), Stilton (3.62 μg per 100 g) [14]. Other tested cheeses contained less than 3 μg per 100 g. Total vitamin K2 was the highest in Münster (80.1 μg per 100 g), Camembert (68.1 μg per 100 g), Gamalost (54.2 μg per 100 g), Stilton (49.4 μg per 100 g), Emmenthal (43.3 μg per 100 g), Norvegia (41.5 μg per 100 g), Roquefort (38.1 μg per 100 g), and Raclette (32.3 μg per 100 g). The rest of examined cheeses comprised less than 3 μg per 100 g of vitamin K2 [14].

### 2.4. Vitamin K in Meat and Fish

Vitamin K is present in meat and fish, although there are inconsistencies in the total content of vitamin K depending on the origin of the meat [15]. For example, the amount of MK-4 differs in chicken meat in the United States (13.6–31.6 μg per 100 g), compared to the Netherlands (5.8–11.3 μg per 100 g), and Japan (27 ± 15 μg per 100 g) [17]. In Europe, vitamin K1 is found in deer back (2.4 μg per 100 g), beef liver (2.3 μg per 100 g), and minced meat (1.1 μg per 100 g) [14]. Values less than 0.5 μg per 100 g were not deemed noteworthy for this review. The richest K1 fish sources are eel (1.3 μg per 100 g) and mackerel (0.5 μg per 100 g) [14].

Total vitamin K2 was present in meaningful concentrations in chicken meat (10.1 μg per 100 g), beef liver (11.2 μg per 100 g), minced meat (7.6 μg per 100 g), beef meat (1.9 μg per 100 g), and pork liver (1.8 μg per 100 g) [14]. Only values above 1.5 μg per 100 g for meat were chosen as representatives. Among the fish tested in this study, eel (63.1 μg per 100 g) had the highest concentration of vitamin K2, followed by plaice (5.3 μg per 100 g), mackerel (0.6 μg per 100 g), and salmon (0.6 μg per 100 g). The rest of the examined fish species contained less than 0.5 μg per 100 g of vitamin K1 and total vitamin K2 [14].

## 3. Adequate Intake is an Estimate

Vitamin K content can vary depending on the approach used for detection, especially in the case of K2. The gold standard for measuring vitamin K is by high-performance liquid chromatography (HPLC) using C-18 reversed phase column and fluorometric detection after post-column zinc reduction. This technique is used frequently to analyze vitamin K content in food and enables quantification of separate vitamin K isoforms. For example, a European standard method (EN 14148:2003) was developed to determine vitamin K1 by HPLC; however, no official registered method exists for K2. Typical recommended vitamin K intake in North America varies from 50 to as high as 600 μg/day for vitamin K1, and from 5 to 600 μg/day for vitamin K2 [12]. The challenge lies in assessing an adequate intake, as current requirement is an estimate based on regional consumption. Currently, daily recommended intake for vitamin K is-based solely on intake of vitamin K1, blood coagulation and differs by region [18,19,20]. It is defined as the median of daily intake in healthy individuals and differs worldwide. According to National Academy of Medicine (NAM), previously known as the Institute of Medicine, the required vitamin K dosage for adult men is 120 µg/day and 90 µg/day for women. According to the World Health Organization (WHO) and the Food and Agriculture Organization (FAO), the dosage is 65 µg/day for men and 55 µg/day for women, on the basis of 1 µg/day/kg vitamin K. The Commission of the European Communities sets a RDI for vitamin K of 75 µg/day [12,21,22,23]. In case of vitamin K1, NAM implies that the average intake is currently already higher than adequate (surpassing 100%) [22]. Therefore, daily recommended intake of vitamin K1 can be achieved easily by the Western diet, as no deficiency in vitamin K1 has been reported so far in healthy adults [12]. However, intake of vitamin K2 from food corresponds to only 25% and 5% of the total vitamin K intake, respectively [21]. Therefore, administration of K2 via supplements in a high dosage might be advisable for meeting the required daily intake for improvement of overall health [12].

## 4. Functions of Vitamin K1 and K2

The most well-known function of vitamin K is as a cofactor in the activation of vitamin K-dependent coagulations factors [24,25]. Through post translational modification of glutamate (Glu) residues in coagulation factors, vitamin K enables posttranslational carboxylation, which allows high affinity binding with support of calcium to negatively charged phospholipid membrane areas and therefore maintains hemostasis [26,27,28].

Due to well-established guidelines, vitamin K1 is also administered as a medication. For instance, newborns are given 1 mg K1 shortly after birth to prevent the potentially lethal vitamin K deficiency bleeding (VKDB). VKDB occurring within the first 24 hours of life is uncommon and is usually caused by drugs prescribed to the mother. These may interfere with fetal vitamin K metabolism [29]. Without prophylaxis, VKDB can occur within the first week after birth due to vitamin K deficiency caused by insufficient placental transfer and low concentrations in breast milk of vitamin K [30,31]. Moreover, vitamin K1 is used as an antagonist in patients on vitamin K Antagonist (VKA) treatment prior to elective surgery or when international normalized ratio (INR) values are too high (prolonged bleeding) [32,33].

Other functions of vitamin K are mainly ascribed to vitamin K2 (discussed below). Different strains of bacteria synthesize a variety of menaquinones, the exception being MK-4. MK-4 is a product of tissue specific conversion from phylloquinone, and can be found in mammals [34]. Nakagawa et al. identified UbiA prenyltransferase containing 1 (UBIAD1), a human homologue of *Escherichia coli* prenyltransferase, responsible for the conversion of phylloquinone to MK-4 in several tissues such as the cerebrum, liver and pancreas [35]. The discovery of a MK-4 biosynthetic enzyme in humans and confirmation that MK-4 originates from phylloquinone might support the rationale behind a lower efficacy of MK-4 compared to other menaquinones.

## 5. Vitamin K: The Vitamin K Cycle

Both vitamin K1 and K2 can function as cofactors in the carboxylation process of vitamin K-dependent proteins (VKDPs). Given this, vitamin K serves as a cofactor for γ-glutamylcarboxylase (GGCX), which catalyzes the Glu residue of VKDPs into γ-carboxyglutamic acid (Gla). This process is driven by the oxidation of vitamin K-hydroquinone (KH_2_) to vitamin K-epoxide (KO) in the vitamin K cycle. The vitamin K-oxidoreductase (VKOR) converts KO to vitamin K and back to KH_2_, generating a recycling process of vitamin K [36,37]. Consequently, VKAs inhibit VKOR and thereby the recycling of vitamin K resulting in a drug-induced vitamin K deficiency [38].

VKDPs are categorized as hepatic and extra-hepatic VKDPs. Hepatic VKDPs include coagulation factors II, VII, IX, X, and anticoagulant protein C, protein S, and protein Z, all of which are involved in regulating blood coagulation. Extra-hepatic VKDPs include Matrix Gla protein (MGP), Osteocalcin, and Gla-rich protein (GRP) [36]. These VKDPs are primarily involved in maintaining bone homeostasis, as well as inhibiting ectopic calcification [39]. MGP is primarily expressed in vascular smooth muscle cells and chondrocytes. This protein is known to inhibit extracellular matrix mineralization of vascular lesions and is involved in vascular remodeling [40,41,42]. Osteocalcin can be defined as a bone tissue-specific protein and is involved in regulating mineral deposition [36,43]. GRP can be found in calcified cartilage and vasculature, where it directly binds and inhibits crystal formation/maturation and vascular smooth muscle cell calcification [36,44,45].

In addition to the role of vitamin K in the carboxylation of VKDPs, recent studies have suggested a role for vitamin K as an antioxidant. A paralogue enzyme of VKORC1, named VKORC1-like 1 (VKORC1L1), is expressed in many tissues. VKORC1L1 has been shown to mediate vitamin K-dependent intracellular antioxidant function in the human cellular membrane [46]. Indeed, vitamin KH_2_ exhibits an antioxidative activity of 10- to 100-fold higher than any other known radical scavengers, such as alpha-tocopherol and ubiquinone [47]. Vitamin K protects the cellular membrane from lipid peroxidation [48]. This antioxidative activity is diminished in the presence of warfarin. Both vitamin K1 and K2 prevent oxidative stress in neuronal cells and primary oligodendrocytes through inhibition of 12-Lipoxygenase [49,50]. In addition to antioxidative properties, vitamin K has been reported to facilitate ATP generation and rescue mitochondrial dysfunction [51].

## 6. Vitamin K: Bioavailability

Of all the menaquinones, MK-7 is absorbed most efficiently and exhibits greatest bioavailability [6,13]. This was shown by a comparative study between K1 and MK-7 following intake of vegetables and K2 rich food, respectively. Both vitamin K1 and MK-7 were readily absorbed within 2 hours after ingestion. However, postprandial serum concentrations of K2 (MK-7) were 10-fold higher than K1 [6]. K1 showed large interindividual variation in fasting plasma concentrations [52]. Vitamin K1 also showed inferior absorption in comparison to MK-4, as well as longer chain menaquinone food sources (MK-8 and MK-9) [6]. Long chain menaquinones, such as MK-7 and MK-9, have longer half-life in circulation in comparison to K1 (Figure 2) [53]. Thus, it is available for longer in the circulation to be absorbed by extrahepatic tissue [13]. However, not all menaquinones are equally well absorbed. The bioavailability of MK-4 does not reflect an increased serum concentration after administration. However, MK-7 administration is reflected in increased serum MK-7 levels up to several days, thereby contributing to vitamin K status [54]. Finally, MK-9, being even more lipophilic, has a very long half-life, but due to the lipophilicity is poorly absorbed [53].

## 7. Vitamin K: Uptake and Distribution

All vitamin K forms can be taken up by enterocytes in the small intestine and packaged into chylomicrons during absorption. These chylomicrons are then taken up by the liver. Moreover, vitamin K1 has a rapid removal rate from the circulation. This was revealed by giving isotope-labeled phylloquinone to a volunteer, which then showed rapid appearance of radiolabeled excretory metabolites in urine and bile [55,56]. Vitamin K1 is preferentially retained in the liver to assist carboxylation of clotting factors [53]. In contrast, vitamin K2, particularly long chain derivatives, are redistributed to the circulation and are available for extra-hepatic tissues such as bone and vasculature (Figure 2) [5,53].

## 8. Vitamin K2 in Health and Disease

Vitamin K2-dependent proteins are expressed in a varying manner in both soft and hard tissues. The role of K1 in coagulation is well established and has been reviewed elsewhere. Whereas vitamin K2’s position in health and disease is well recognized within CVD, bone development and fractures, chronic kidney disease and certain cancers. Additional evidence is demonstrating further roles of vitamin K2 in liver disease, immune function, neurological diseases and obesity.

### 8.1. Vitamin K2: Cardiovascular Disease

Vascular calcification is an active process that causes CVD, the largest killer in the world [57]. It is well known that vitamin K2-dependent proteins activate a protective mechanism preventing the development of vascular calcification [58]. Furthermore, vitamin K2, in the form of MK-7, has been proven in numerous trials with healthy and diseased patient cohorts to have a long-term protective effect on the development of calcification [59,60,61,62]. Additionally, several studies have demonstrated an overall reduction in risk of CVD development [63]. Even a regression of arterial stiffening and improvement of vascular elasticity have been observed in healthy population cohorts following supplementation. Interestingly, in a study investigating all vitamin K1 and K2 isoforms, only K2 was effective and beneficial for CVD health and not K1 [64]. Furthermore, the role of K2 in CVD is profound and detailed discussion can be found elsewhere [58,59,65]. At present, there are multiple large clinical studies being performed around the world with vitamin K2 supplementation in a variety of cardiovascular patients, the results of which will add further substance to the role of vitamin K2 in CVD [66].

### 8.2. Vitamin K2: Bone Fractures and Degeneration

Bone fracture and quality is important in any aged population. Vitamin K2 is known to improve bone quality, which in turn reduces fracture risk as demonstrated by numerous studies with population groups above the age of 50 [67,68,69,70]. Furthermore, children are inherently born vitamin K2 deficient, and there is the implication that should this not be corrected, inadequate bone formation might take place [71]. This is substantiated by mutations to vitamin K-dependent enzymes, resulting in birth defects that affect development of bone and cartilage [72]. Within bone marrow mesenchymal stem cells, vitamin K2 treatment supports osteogenic differentiation [73]. Osteocalcin expression in bone is related to proper function, although the precise mechanism is under ongoing research [58]. Further to this, there are population studies currently underway that will shed further light on the role of vitamin K2 on bone development, health and maintenance [74,75].

### 8.3. Vitamin K2: Diabetes Mellitus

Long-term supplementation of vitamin K2 has been shown to reduce the risk of diabetes development. The largest study, with 38,000 men and women, aged 20–70, demonstrated that just 10 μg/day of K2 decreases diabetes risk by 7% [76]. The mechanism by which this K2 may act in doing so is beginning to be unraveled. Vitamin K2 activates osteocalcin, which has been shown in vitro to promote proliferation of pancreatic beta cells as well as increasing insulin production and expression of CyclinD1 [77,78,79]. The specific mechanism is currently under research and it is hypothesized that osteocalcin, lectin and adiponectin have an intricate network for glucose metabolism that can be modulated by vitamin K2 [80].

### 8.4. Vitamin K2: Cancer

Interestingly, vitamin K2 has been explored in several clinical interventions to supplement cancer treatments [81]. In vitro studies found K2 supplementation alone to prevent growth and metastasis of multiple cancer cell lines [82,83,84]. The mechanisms by which vitamin K2 can inhibit proliferation and metastasis of cancers has been reviewed elsewhere [80]. In short, vitamin K2 may act in several pathways including protein kinase A, protein kinase C, nuclear factor kappa B and steroid and xenobiotic receptor [81,85]. Furthermore, there are multiple cases by which K2 supplementation alongside standard treatment subsided cancer development, including multiple cases where patients entered complete remission [86,87]. Remarkably vitamin K2’s action as an anticancer agent is not limited to a definitive cancer type, and instead has been reported in multiple cancer forms [81].

### 8.5. Vitamin K2: Liver Disease

The role of vitamin K in the liver has been well established with regards to production of coagulation factors and activation of VKDPs [88,89]. While the majority of research has focused on K1, K2 has a higher bioactivity, and might act in a similar manner to hepatic tissue. Additionally, emerging vitamin K2 research is demonstrating a regenerative effect on oval cells, as well as maturation of hepatic cells from stem cell cultures [90,91]. This suggests that there might be a developmental importance of K2 in the liver. Furthermore, within the anticancer effects of vitamin K2 several trials have found MK-4 to be an effective agent against hepato-carcinomas that have arisen from alcoholic and non-alcoholic liver cirrhosis [85,92]. Moreover, there is a positive trend in vitamin K2 supplementation in treating liver cirrhosis, although further investigation is required to determine whether this is significant and to understand the mechanism by which vitamin K2 acts in liver disease.

### 8.6. Vitamin K2: Chronic Kidney Disease

Status of dephosphorylated-uncarboxylated-MGP (dp-ucMGP) is an accepted research marker for vitamin K deficiency first described in patients with chronic kidney disease [93]. Dp-ucMGP is associated with progression of CKD, being that later-stage CKD patients have higher circulating dp-ucMGP levels [94,95,96,97]. Vitamin K2 supplementation has been shown to improve renal artery function and prevent further development of renal artery calcification [62,98]. This is of overall benefit to renal function. Further to this, it has been demonstrated that vitamin K2 supplementation may improve glomerular filtration in a patient cohort [96]. The promise of vitamin K2 in CKD research is great and there are multiple large-scale studies underway using vitamin K supplementation to treat patients with CKD [74,99,100,101].

### 8.7. Vitamin K2: The Immune System

In recent years, ex vivo studies have demonstrated a previously unknown immunomodulatory role for vitamin K2. First, it was demonstrated that MK-7 modulated expression of TNF-α, IL-1α and IL-1β [102]. Furthering this finding, K2 decreases proliferation of T-cells from healthy individuals, whereas vitamin K1 had no such effect [103]. This has been further substantiated with T-cells from a larger number of children with pediatric atopic dermatitis and healthy controls, as well as a separate study with patients on dialysis [104,105]. Both these studies demonstrated that K2 decreased the number of activated T-cells, as well as proliferation. Thus, accumulated evidence is showing a novel role of K2 as an immunosuppressive agent. This needs to be further elaborated; until then, a novel physiological mechanism by which vitamin K2 can aid immunomodulation can be hypothesized, although it requires further study.

### 8.8. Vitamin K2: Neurological Disease

Extrahepatic activity of vitamin K2 has been elucidated in vivo by varying activity of K2 recycling enzymes in different tissues. Prominently, a subset of vitamin K2 enzymes have been shown to be highly expressed in the brain [106]. A protective effect of K2 on neurons in vitro has been documented [107]. MK-4 improved energy production and rescued PINK1 mutation found in Parkinson’s disease [51]. More recently, research identified vitamin K2 protection of neurons via a novel mechanism involving P38 MAP kinase pathway [108]. Further to this, various K2 analogs have been found to be decisive in neuronal differentiation [109]. The first epidemiological study of K2 in relation to neuronal activity involved a cohort of 45 patients with multiple sclerosis (MS) and 29 healthy volunteers [110]. K2 levels were greatly reduced in patients with MS compared to the controls that were gender and age dependent. Also, K2 levels were correlated with neurological spasms and lesions of the optic nerves. These emerging studies suggest a potentially important role for vitamin K2 in neurological development and disease.

### 8.9. Vitamin K2: Obesity

The link between osteocalcin and adiponectin has been strongly suggested; however, the mode by which any such action takes place remains elusive [78,111,112]. Direct elevation of uncarboxylated osteocalcin (ucOC) levels reduced fat mass and improved glucose metabolism in mice [79,113]. More recently, supplementation of vitamin K1 and MK-4 in rats reduced total fat and serum triglyceride levels [114]. Further to this, a mechanism was implicated by Ding et al., whereby VKORC1L1 was found to promote adipogenesis [115]. Consequently, downregulation of VKORC1L1 increased intracellular K2 levels, and preadipocyte differentiation was inhibited [115]. Human cohort studies have shown improvements in body weight, waist circumference, body composition, visceral fat and diabetes mellitus from K2 supplementation [116,117,118,119]. This all suggests an overall beneficial effect of vitamin K2 on glucose and fat metabolism that requires further investigation to confirm.

## 9. Conclusions and Future Perspective

The existence of vitamin K has been known for over 80 years, namely via its essential role in coagulation. The discovery of different isoforms of vitamin K is beginning to elucidate a significant role of vitamin K outside of coagulation. Functions of K2 are proving to be beneficial with regard to CVD and bone metabolism. There is a growing body of evidence suggesting vitamin K2 is involved in multiple cellular processes, and might have a protective role in various organs throughout the human body (Figure 3). While vitamin K2 has improved outcomes in many clinical trials, the exact mechanism of action remains to be unraveled. Major health organizations, such as WHO, European Food Safety Authority (EFSA) and Food and Drug Administration (FDA) have established RDI for vitamin K, which is solely based on the dose of K1 to retain an appropriate blood clotting function. This review highlights and summarizes differences between vitamin K1 and K2 in intake and function. When exploring the non-coagulation, extrahepatic activity of vitamin K, it is clear that K2 in its various forms is the highlight of such activity. Therefore, although history and nomenclature have classed K1 and K2 into the same category, these molecules can have a very different action in the body. Though a new realm to vitamin K seems to be on the horizon, whether it opens up new answers in health and disease remains to be seen. Differences between K1 and K2 merit recognition among national and international regulatory organizations, and remain open to research.

## Figures and Tables

**Figure 1 ijms-20-00896-f001:**
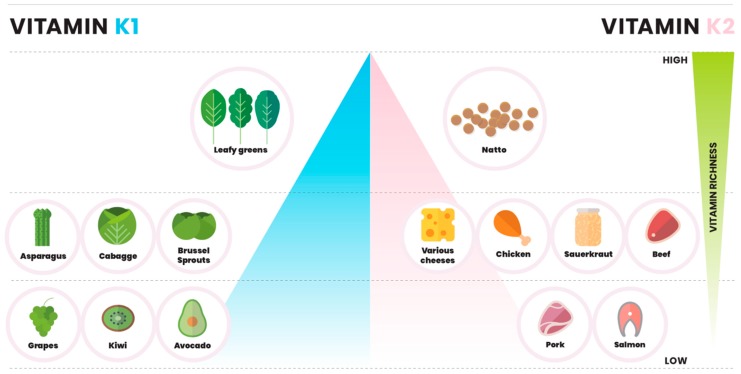
Dietary sources of vitamin K. Left side of pyramid displays K1 content gradient in dietary sources of vitamin K1. Leafy greens include spinach, kale and swiss shards. Right side visualizes K2 content gradient with natto being the most significant source. Various cheeses include hard and soft cheeses with K2 content being dependent on fermentation level.

**Figure 2 ijms-20-00896-f002:**
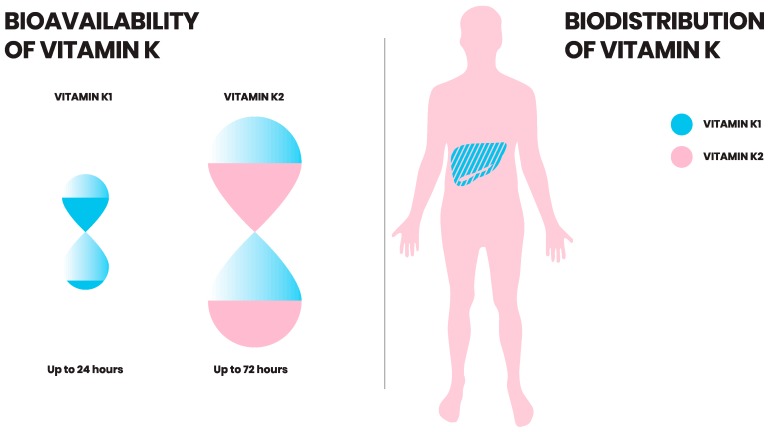
Bioavailability and biodistribution of vitamin K. Vitamin K2 has a longer half-life in the circulation than vitamin K1. While vitamin K1 is retained and exerts its function in the liver, vitamin K2 is redistributed to the circulation and (extra-)hepatic tissues.

**Figure 3 ijms-20-00896-f003:**
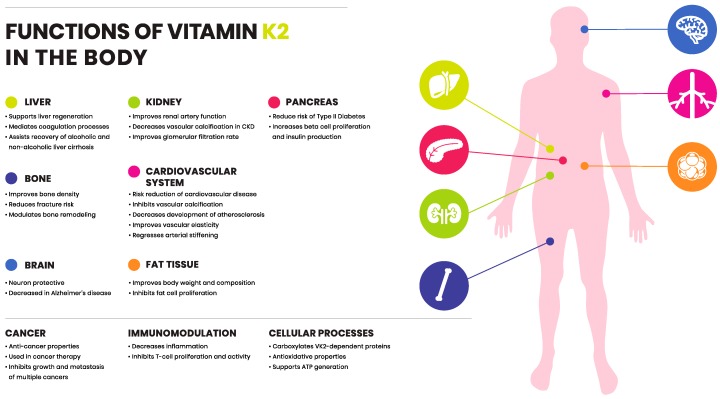
Functions of vitamin K2 in the body. Vitamin K2 exerts protective role and is involved in various organ systems throughout the human body (summarized in the figure).

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
