# Peer review of "Vitamin K: Double Bonds beyond Coagulation Insights into Differences between Vitamin K1 and K2 in Health and Disease"

_ijms, 2019, doi:10.3390/ijms20040896_

Round 1
Reviewer 1 Report
The review article is well written and summarizes the recent research of Vitamin K2 in extra-hepatic tissues. There is no comparable Review available until now.
I have just some minor comments:
- Is it possible to change the units in the section "Vitamin K in vegetables in µg/100g? Or please recalculate everything "per cup" (row 66-71)
- In the section "Vitamin K in fruits and nuts" (row 72-80) the units are in "per 100g" and in the following section (row 81-93) "per g". Would be nice to have everything in the same unit to have a direct comparison
- in row 173 ther is written that VKORC1L1 is a subunit of VKORC1 which is not the case. VKORC1L1 is the paraloge Enzyme of VKORC1!
- row 224 first time used the Abbreviation CVD without Explanation. inlude in row 216 the Abbreviation
Author Response
The review article is well written and summarizes the recent research of Vitamin K2 in extra-hepatic tissues. There is no comparable Review available until now.
Response: We thank the reviewer for the kind words and a favourable response to our review.
I have just some minor comments:
1. Is it possible to change the units in the section "Vitamin K in vegetables in µg/100g? Or please recalculate everything "per cup" (row 66-71)
Response: We have recalculated all the measurements into µg per 100 g and used the calculation of 1/2cup =75 g and 1cup=150g.
2. In the section "Vitamin K in fruits and nuts" (row 72-80) the units are in "per 100 g” and in the following section (row 81-93) “per g”. Would be nice to have everything in the same unit to have a direct comparison
Response: We have changed the units and unified them.
3. In row 173 there is written that VKORC1L1 is a subunit of VKORC1 which is not the case. VKORC1L1 is the paraloge Enzyme of VKORC1!
Response: We thank the reviewer for pointing out the incorrect description in the review. We removed the sentence and gave a new definition to VKORC1L1. It reads now:
“A paralogue enzyme of VKORC1, named VKORC1-like 1 (VKORC1L1), is expressed in many tissues”
4. row 224 first time used the Abbreviation CVD without Explanation. include in row 216 the Abbreviation
Response: We have now amended this throughout the review.
Reviewer 2 Report
In this review, authors aim at describing the different functions of vitamin K1 and K2, in both physiological and pathological contexts. They especially highlight a main problem regarding health organizations about vitamin K required intake : It differs a lot depending on geographical regions, and is only based on vitamin K1 (and thus coagulation), whereas vitamin K2 is involved in many others processes and needs to be also considered.
First of all, figures are really nice and easy to understand, making interpretation of vitamin K sources, distribution and functions highly clear.
However, informations in the text part considering vitamin K in vegetables, fruits and nuts, cheese, meat and fish is not always so easy to read. Maybe the authors wrote too many concentrations, sometimes with different units, which makes interpretation a bit unclear. Authors might give only a ranking of the different aliments depending on their vitamin K concentration, or might show these informations as graphical.
Moreover, some questions remain regarding the differences between vitamin K1 and K2 : Is vitamin K1 involved in other processes than coagulation ? Is it able to support other mechanisms beside vitamin K2 ? Authors also indicate that MK-4 can be formed from vitamin K1, but are there any informations about K1, MK-4, MK-7 or MK-9 metabolic interactions ? Finally, roles of MK-4 and MK-7 are cleary identified, but is some data available about MK-9 physiological function(s) ?
But the most important point is about VKORC1-Like (VKORC1L1), that authors defined as a « VKORC1 subunit ». What do the authors mean by « subunit » ? Because VKORC1L1 appears to be a VKORC1 paralog enzyme, not a subunit.
Nevertheless, this is a high quality review, bringing an overview of vitamin K functions – beside its role in coagulation, already widely reported. It mainly highlights the diversity of vitmain K2 biological functions. In this way, authors bring out the necessity to carry on research about vitamin K metabolism, and the necessity for health organizations to consider vitamin K2 as well as vitamin K1 for intake recommendations and for future therapies.
Author Response
In this review, authors aim at describing the different functions of vitamin K1 and K2, in both physiological and pathological contexts. They especially highlight a main problem regarding health organizations about vitamin K required intake: It differs a lot depending on geographical regions and is only based on vitamin K1 (and thus coagulation), whereas vitamin K2 is involved in many others processes and needs to be also considered.
First of all, figures are really nice and easy to understand, making interpretation of vitamin K sources, distribution and functions highly clear.
Response: We thank the reviewer for a favourable response to the manuscript and the figures
However, information in the text part considering vitamin K in vegetables, fruits and nuts, cheese, meat and fish is not always so easy to read. Maybe the authors wrote too many concentrations, sometimes with different units, which makes interpretation a bit unclear. Authors might give only a ranking of the different aliments depending on their vitamin K concentration, or might show these informations as graphical.
Response: We adjusted all the units to µg per 100 g enabling uniform format
Moreover, some questions remain regarding the differences between vitamin K1 and K2 : Is vitamin K1 involved in other processes than coagulation ?
Response:
Vitamin K is an unequivocal cofactor to support carboxylation VKDP, which is found in hepatic and extrahepatic tissues. Therefore, vitamin K is involved in other processes than coagulation only. It has been shown that also vitamin K1 is involved in bone and vascular tissue. However, vitamin K1 is preferentially retained in the liver and has shorter bioavailability as compared to long-chain vitamin K2. Hence, the involvement of vitamin K1 is predominantly related to the hepatic system and hence coagulation. We addressed the physiology of vitamin K1 in the “Vitamin K: Uptake and distribution section”:
“Moreover, vitamin K1 has a rapid removal rate from the circulation.”
“Vitamin K1 is preferentially retained in the liver to assist carboxylation of clotting factors (53). In contrast, vitamin K2, particularly long chain derivatives, are redistributed to the circulation and are available for extra-hepatic tissues such as bone and vasculature”
Is it able to support other mechanisms beside vitamin K2?
Response : “section Vitamin K: The vitamin K cycle”
“Both vitamin K1 and K2 function as cofactor in the carboxylation process of vitamin K dependent proteins (VKDPs).”
“Both vitamin K1 and K2 prevent oxidative stress in neuronal cells and primary oligodendrocytes through inhibition of 12-Lipoxygenase”
Authors also indicate that MK-4 can be formed from vitamin K1, but are there any information about K1, MK-4, MK-7 or MK-9 metabolic interactions ?
Response
The reviewer raises an interesting topic, however thus far, metabolic interaction between MKs has not been studied.
Finally, roles of MK-4 and MK-7 are clearly identified, but is some data available about MK-9 physiological function(s)?
Response: we included the following sentence to indicate the role of MK-9 in absorption.
“Finally, MK-9 being even more lipophilic has a very long half-life but due to the lipophilicity is poorly absorbed.”
But the most important point is about VKORC1-Like (VKORC1L1), that authors defined as a « VKORC1 subunit ». What do the authors mean by « subunit » ? Because VKORC1L1 appears to be a VKORC1 paralog enzyme, not a subunit.
Response: We thank the reviewer for pointing out the incorrect description in the review. We removed the sentence and gave a new definition to VKORC1L1. It has been adapted to:
“A paralogue enzyme of VKORC1, named VKORC1-like 1 (VKORC1L1), is expressed in many tissues”
Nevertheless, this is a high quality review, bringing an overview of vitamin K functions – beside its role in coagulation, already widely reported. It mainly highlights the diversity of vitamin K2 biological functions. In this way, authors bring out the necessity to carry on research about vitamin K metabolism, and the necessity for health organizations to consider vitamin K2 as well as vitamin K1 for intake recommendations and for future therapies.
Response: We thank the reviewer for the kind words and a favourable response to our review.